# The Crosstalk of miRNA and Oxidative Stress in the Liver: From Physiology to Pathology and Clinical Implications

**DOI:** 10.3390/ijms20215266

**Published:** 2019-10-23

**Authors:** Eckhard Klieser, Christian Mayr, Tobias Kiesslich, Till Wissniowski, Pietro Di Fazio, Daniel Neureiter, Matthias Ocker

**Affiliations:** 1Institute of Pathology, Paracelsus Medical University/Salzburger Landeskliniken (SALK), 5020 Salzburg, Austria; d.neureiter@salk.at; 2Cancer Cluster Salzburg, 5020 Salzburg, Austria; 3Department of Internal Medicine I, Paracelsus Medical University/Salzburger Landeskliniken (SALK), 5020 Salzburg, Austria; christian.mayr@pmu.ac.at (C.M.); t.kiesslich@salk.at (T.K.); 4Institute of Physiology and Pathophysiology, Paracelsus Medical University/Salzburger Landeskliniken (SALK), 5020 Salzburg, Austria; 5Department of Gastroenterology and Endocrinology, Philipps University Marburg, 35043 Marburg, Germany; wissniowski@me.com; 6Department of Visceral, Thoracic and Vascular Surgery, Philipps University Marburg, 35043 Marburg, Germany; difazio@med.uni-marburg.de; 7Translational Medicine Oncology, Bayer AG, 13353 Berlin, Germany; Matthias.Ocker@bayer.com or; 8Department of Gastroenterology CBF, Charité University Medicine Berlin, 12200 Berlin, Germany

**Keywords:** microRNA, oxidative stress, metabolism, physiology, ASH, NAFLD, NASH, HCC, HCV, HBV

## Abstract

The liver is the central metabolic organ of mammals. In humans, most diseases of the liver are primarily caused by an unhealthy lifestyle–high fat diet, drug and alcohol consumption- or due to infections and exposure to toxic substances like aflatoxin or other environmental factors. All these noxae cause changes in the metabolism of functional cells in the liver. In this literature review we focus on the changes at the miRNA level, the formation and impact of reactive oxygen species and the crosstalk between those factors. Both, miRNAs and oxidative stress are involved in the multifactorial development and progression of acute and chronic liver diseases, as well as in viral hepatitis and carcinogenesis, by influencing numerous signaling and metabolic pathways. Furthermore, expression patterns of miRNAs and antioxidants can be used for biomonitoring the course of disease and show potential to serve as possible therapeutic targets.

## 1. Introduction

The liver is the central metabolic organ in the human body. It serves as a storage organ for e.g., glycogen, lipoproteins, vitamins, iron and blood, synthesizes important proteins such as albumins, transferrin and coagulation factors as well as fats and lipoproteins. Enzyme systems that are necessary for the metabolism of fat–for example–are involved in the formation of reactive oxygen species (ROS) and reactive nitrogen species (RNS), which in turn play a role in the development of non-tumours and tumorous liver diseases like non-alcoholic fatty liver disease (NAFLD) and liver cancer (hepatocellular carcinoma (HCC)).

In the human body there is normally a balance between antioxidants and ROS. If, however, the metabolic situation changes in favor of ROS, then oxidative stress (OS) is present in the cell [1]. ROS are almost always considered to be purely toxic, although ROS also have important regulatory tasks in various signaling pathways [2,3,4]. After growth factor stimulation an increase of ROS is necessary for the increase of tyrosine phosphorylation, which is needed for downstream signaling [5,6]. In the bone marrow, ROS play an important role in the redox regulation of stem cells and progenitor cells of hematopoiesis [7]. Besides these beneficial tasks, ROS are involved in aging and carcinogenesis, although they also play an ambivalent role herein [8,9,10,11]. This also applies to antioxidants which degrade ROS and thus ensure the stability of the genome but that also prevent apoptosis of damaged cells [12]. Aging and carcinogenesis are also part of the spectrum of activity of micro-RNAs (miRNAs) [13,14,15,16]. miRNAs do not code for proteins but play a role in the regulation of the expression of genes that are involved in regulation of diverse biological pathways. miRNAscause either degradation (if they match perfectly to messenger RNA) or prevention of translation (imperfect match) of the respective mRNAs [17]. The main task of miRNAs is gene regulation. They are a pillar of self-regulation, but also interact with other mechanisms of epigenetics such as histone modification [18]. Inferred from this, they have a decisive role not only in malignant diseases, but also in physiologic conditions and in metabolic diseases like NAFLD [19]. This review is intended to provide a comprehensive overview of the (inter)actions of oxidative stress and miRNA in pathological processes of the liver.

## 2. Physiology and Metabolism

In the context of oxidative stress, ROS are a group of chemically reactive, intracellular compounds containing oxygen and include the superoxide anion (O_2_^−^), hydrogen peroxide (H_2_O_2_) and the hydroxyl radical (HO•)–each with different chemical properties such as reactivity, half-life, diffusion distance, and permeability through cellular membranes [20,21,22]. While these species have physiologic functions in cell signaling and regulation (“oxidative eustress”), supraphysiologic oxidative levels may cause damage to biomolecules and cells, i.e., “oxidative distress”. The general concept of oxidative stress is defined as a dysbalance favoring oxidants (ROS as well as RNS) over antioxidants thus disrupting redox signaling and control and/or inducing molecular damage [23,24] (see Figure 1).

In non-phagocytic cells, mitochondria represent the main sources of ROS produced within the steps of oxidative phosphorylation. Catalyzed by NADP(H) or xanthine oxidase, about 1% of the mitochondrial electron flow contributes to generation of superoxide anion. Importantly, at physiologic levels, free radicals play a role in the cell’s signal transduction [25], regulation of gene expression and defense against pathogens [1]. As reviewed by Dickinson and Chang, other cellular sources of ROS include the endoplasmic reticulum during oxidative protein folding mechanisms (post-translational protein disulfide bond formation) and NADPH oxidases (NOX) located at various cellular membranes [26]. In the context of an immune response, NADPH-dependent enzymes such as NOX2 seem indispensable [27]. In the gut, bacteria stimulate ROS production via NOX1 and DUOX2 and ROS promote intestinal stem cell proliferation [12,27,28,29].

Besides side reactions in the electron flow within the oxidative phosphorylation pathway, (ethanol-inducible) cytochrome P450 enzymes (CYP2E1) represent a non-mitochondrial source of ROS in the liver [1,30]. As reviewed by Li et al. [31], other sources of ROS include hepatic metabolization of drugs, environmental pollutants and other factors such as radiation, temperature, high fat or high salt diet. As a tissue characterized by high metabolic activity, the liver parenchyma is equipped with several ROS scavenging mechanisms: besides non-enzymatic factors (α-tocopherol, glutathione (GSH), β-carotene, bilirubin, flavonoids, and plasma proteins [22]), the nuclear factor erythroid 2 like 2 (Nrf2) is a cellular redox sensor which—induced by elevated levels of ROS—is released from sequestration via the cytoplasmic cytoskeletal-anchoring protein Kelch-like ECH-associated protein 1 (Keap1) and, in turn, promotes transcription of ROS-protective genes. Genes regulated by Nrf2 via antioxidant response elements (ARE) include ROS-relevant factors involved in GSH turnover (regeneration), reduction of oxidized protein thiol groups and NADPH-producing enzymes (required for drug-metabolizing enzymes and antioxidant systems)–for review see Hayes and Dinkova-Kostova [32]. In the context of hepatic pathology, Nrf2-mediated cytoprotective responses are involved in (counteracting) the development of various liver diseases including alcoholic and non-alcoholic liver diseases, viral hepatitis, fibrosis and HCC. Therefore, ROS are central factors in the pathogenesis of various hepatic diseases [1]–as summarized in Table 1.

Most chronic liver diseases are characterized by deposition and accumulation of extracellular matrix components (collagens, fibronectin, elastin, laminin, hyaluronan, and proteoglycans), mostly secreted by hepatic stellate cells (HSC), resulting in up to six times more extracellular matrix (ECM) than normal in advanced stages of fibrosis [33]. Depending on genetic and environmental factors, fibro-proliferative disorders (i.e., NAFLD or NASH) may proceed to liver cirrhosis, which in its uncompensated form, is associated with acute and chronic liver failure, portal hypertension and often require liver transplantation [34]. The involvement of ROS during the development of liver fibrosis is evident based on several mechanisms [23] (see also [22,35,36] for reviews): i) ROS-based stimulation of collagen (Col1α1) synthesis by HSC, ii) ROS-based intracellular signaling of transforming growth factor β (TGFβ) as a major fibrogenic factor as well as its up-regulation by ROS, and, iii) ROS produced by NOX enzymes contributing to HSC activation.

Taken together, production of ROS and correlated tissue damage represent central aspects of various hepatic diseases. Therefore, understanding (epigenetic) regulation of factors involved in either generation of ROS or in their detoxification is necessary to fully comprehend pathophysiologic mechanisms of liver diseases as well as to develop new epigenetics-based therapeutic approaches.

## 3. Alcoholic Liver Disease (ALD) and Alcoholic Steatohepatitis (ASH)

ASH is a liver disease caused by high alcohol consumption. The accumulation of ethanol and its metabolic products lead to production of ROS that alter the hepatocyte function, finally leading to fibrosis, cirrhosis, and in 5% to 6% of patients, to the development of HCC. Beyond the genetic and metabolic alterations occurring during ASH, epigenetic modifications have been shown to exert a key role. Changes in DNA methylation at the promoter regions of several genes were discovered in ASH, as were changes in histone acetylation. Nonetheless, it has been found that also miRNAs are differentially expressed in patients affected by ASH [37].

Alcohol intake favors the hepatic accumulation of lipopolysaccharide (LPS), a bacterial antigen, thus mediating the activation of Toll Like Receptor 4 (TLR4). This promotes the transcriptional activity of Nuclear Factor kappa B (NFκΒ), leading to the expression of miR-155. The over-expression of miR-155 causes the release of tumor necrosis factor α (TNFα), ROS and oxidative stress in Kupffer cells (liver resident macrophages) and HSC [38,39]. miR-155 exerts a significant role in hepatocytes by suppressing peroxisome proliferator activated receptor α (PPARα). The down-regulation of this anti-oxidative enzyme causes the over-expression of genes involved in lipid metabolism and uptake, e.g., Fatty Acid Binding Protein 4 (FABP4), Acetyl-CoA-carboxylase 1 (ACC1) and Low-density Lipoprotein Receptor (LDLR) [40].

miR-181b-3p has also been found to be responsible for glucose and lipid homeostasis alterations as well as for liver injury and LPS-induced TLR4/NFκΒ activation in murine Kupffer cells [41]. Additionally, miR-291b expression is responsible for the suppression of Toll interacting protein (Tollip) in Kupffer cells, which is a down-regulator of the TLR4/NFκB pathway [42].

miR-34a, a member of the miR-34 family with known tumor suppressor activity because of its ability to promote p53-mediated apoptosis [43], has been found to correlate with alcoholic liver disease by targeting Sirtuin 1 (SIRT1) mRNA and inhibiting its protein coding [44]. Moreover, SIRT1 is a target of miR-217 during alcohol-associated inflammation [45]. Mice with hepatic deletion of miR-122 develop steatosis at birth, leading to fibrosis and HCC. Its expression is strongly down-regulated in alcohol fed mice as well as in patients affected by alcohol related cirrhosis [46]. Additionally, miR-122 down-regulation enables Hypoxia Inducible Factor 1 alpha (HIF1α) expression in ALD, which contributes to the development of hepatobiliary cancer [47]. It has also been reported that alcohol intake increases the level of miR-155 in Kupffer cells, triggering their sensitization to LPS produced by gut microbiota [48].

Alcohol consumption enhances the level of miR-21, which is overexpressed in several solid tumors including HCC, in hepatocytes and stellate cells [49]. However, its over-expression reduced ethanol-induced cell death, highlighting its role to protect the liver cells during injury [50]. miR-223 is responsible for the peripheral neutrophils activation and liver infiltration induced by ethanol. An increase in its level has been found in serum and neutrophils of patients with elevated alcohol consumption. Its over-expression could trigger NADPH oxidase, thus causing ROS production and liver cell death [51]. Alcohol is furthermore responsible for the suppression of miR-199 in human endothelial cells that leads to steatohepatitis in patients affected by cirrhosis by inducing HIF1α and endothelin-1 (ET-1) [52].

Alcohol-mediated miR-214 expression suppresses cytochrome P450 oxidoreductase (POR), CYP2E1 and glutathione reductase (GSR), which results in oxidative stress in the liver [53] and impairs alcohol metabolism [54]. Table 2 gives a short summary of the ASH associated miRNA and their relation to OS.

## 4. Nonalcoholic Fatty Liver Disease (NAFLD) and Nonalcoholic Steatohepatitis (NASH)

NAFLD is defined by fatty degeneration of hepatocytes comprising more than 5 to 10% of the liver and insulin resistance (IR) but without any history of alcohol abuse and/or other diseases that might lead to fatty liver disease [55]. One third of NAFLD patients progresses to nonalcoholic steatohepatitis (NASH) and fibrosis within 4 to 5 years, depending on the spectrum of lipotoxicity, cellular stress and inflammation [56,57]. NAFLD is caused by an imbalance of free fatty acid (FFA) uptake and *de novo* lipogenesis as well as fatty acid (FA) oxidation and formation of lipoproteins [58,59]. Oxidative stress is seen to be an important player leading to defective hepatocyte regeneration, development of NAFLD and progression to NASH [60]. Excessive nutrients intake, especially high fat diet, leads to excessive FA oxidation [61] and consequently to excessive generation of ROS that are either directly toxic or indirectly by depleting antioxidant reserves [60]. ROS can damage mitochondria, which leads to reduced FA oxidation and accumulation of FA, finally leading to lipotoxicity and release of proapoptotic factors [62]. In turn, lipotoxicity induces endoplasmic reticulum (ER) stress, impairs autophagy and promotes a sterile inflammatory response that aggravates liver cell injury and leads to death of liver cells [63]. Subsequently the unfolded protein response (UPR) is activated by toxic free cholesterol, FFA and diacylglyceride and induces upregulation of proapoptotic C/EBP homologous protein (CHOP) [64,65,66,67]. Usually, UPR induces antioxidant mechanisms by activation of Nrf2 via upregulation of ATF4 transcription factors to counteract the oxidative stress [32]. However, in contrast to normal physiology, NAFLD-related Nrf2 activity is impaired, which also leads to mitochondrial dysfunction and increased intracellular FFA [68,69,70].

Aberrant miRNA expression profiles have been shown to contribute to the development of metabolic syndrome and NAFLD [71]. As also many other genes and pathways that contribute to NAFLD and the progression to NASH are influenced by miRNA, we only provide an overview of the most relevant miRNAs.

miR-21 positively correlates with NAFLD and NASH severity [72]. In hepatocytes, unsaturated FFA increase miR-21 in a mTOR/NFκB dependent manner and inhibit phosphatase and tensin homolog (PTEN) that usually controls FA oxidation in the liver and stimulates glucose uptake in muscle cells [73]. Dattaroy et al. described in 2015 that in HSC NOX upregulates the levels of miR-21, which targets the TGFβ pathway and in turn causes activation of HSC and promotion of fibrogenesis via alpha-1 type I collagen (Col1α1) and alpha smooth muscle actin (α-SMA) upregulation [72]. In 2017 Rodrigues et al. were able to show that ablation of miR-21 results in a progressive decrease in steatosis, inflammation and lipoapoptosis with impaired fibrosis [74]. Fast food diet leads to increased miR-21 levels in liver and muscle of NASH mouse models with concomitantly decreased expression of PPARα, thereby promoting steatohepatitis [74,75].

The best characterized miRNA is miR-122 [76,77,78]. In cases of hepatocellular damage, miR-122 is secreted by damaged cells [79] and appears elevated in the serum during NAFLD. This correlates with disease severity [80,81], although it is contemporaneously reduced in liver tissue [78]. In the context of fibrogenesis, the protective actions of miR-122 are inhibited, which is mediated by long non-coding RNA Nuclear Enriched Abundant Transcript 1 (NEAT1) or via circRNA_002581 and subsequently triggers an increased expression of Kruppel-like factor 6 (KLF6) in HSC [82]. In addition, a miR-122 knockout leads to a higher accumulation of triglycerides (TG), micro steatosis, NASH and fibrosis [83].

Another miRNA that is upregulated in liver tissue and serum and which is integrated into the lipid metabolism is miR-34a [84,85]. Its targets are the transcription factors hepatocyte nuclear factor 4 alpha (HNF4α), PPARα, SIRT1 and p53, all in all leading to an accumulation of TG [86,87,88,89]. miR-34a inhibits SIRT1, which causes the inactivation of AMP-Kinase. This mechanism leads to an increase of hepatic cholesterol synthesis and activation of pro-apoptotic genes (p53 and P66SHC), which contributes to oxidative stress and apoptosis due to reduced β-oxidation resulting in restoration of nicotinamide phosphoribosyltransferase/nicotinamide-adenine-dinucleotide (NAMPT/NAD+) levels and therefore ameliorates hepatic steatosis and inflammation [86,88,90,91].

It was shown that miR-29 family (a, b, c) expression is altered in mice with liver fibrosis and in liver tissue of NASH patients [92,93]. miR-29a and c are downregulated in dietary induced NASH that is accompanied by an upregulation of HMG-CoA reductase (HMCGR), which in turn triggers severe hepatic steatosis and inflammation, probably via enhanced expression of lipoprotein lipase [94,95]. In contrast to that, Kurtz et al. demonstrated that blocking of miR-29 leads to significantly decreased plasma cholesterol and TG levels as a result of the inhibition of *de novo* hepatic lipid synthesis [96]. The reason for these contrary findings could be clarified by Mattis et al. who induced a conditional knockout mouse model and investigated the function of miR-29a [94,95] while Kurtz et al. used the LNA-29 inhibitor to deplete the entire miR-29 family [96]. Furthermore, miR-29b is downregulated in activated mouse HSC, leading to a loss of interaction with Col1α 3’-UTR, which stimulates the collagen production [93,97].

miR-155 is upregulated in a NASH mouse model induced via high fat diet [98]. iR-155 elevates the Forkhead-Box-Protein O3 (FOXO3a) expression thereby regulating the activation of that pathway, whose proteins are involved in the maintenance of the intercellular redox balance [99]. Additionally, miR-155 regulates lipid metabolism by modulating the protein expression of SREBP-1c and fatty acid synthase (FAS) resulting in increased intracellular lipid accumulation in hepatocytes [100]. Interestingly, decreased levels of miR-155 were shown by Csak et al. to be associated with fibrosis via dysregulation of HIF1α and vimentin [101]. This working group showed that a miR-155 knockout reduced steatosis and fibrosis in a mouse model fed with methionine-choline-deficient diet. This leads to the conclusion that miR-155 expression might be stage relevant. In high fat fed mice, miR-155 might exert a protective feedback regulation of the SERBP-1 pathway in order to suppress *de novo* lipid synthesis and reduce lipid load in the hepatocytes [102]. Furthermore, it has been shown that seven miRNAs belonging to the miRNA cluster located at chromosome locus 14q32.2 maternally imprinted region are over-expressed in a NASH mouse model, which was characterized by genetic modification (leptin knock-out) and high fat diet. Therefore, they could represent valid biomarkers for NAFLD/NASH [103].

Many other miRNAs can be linked more directly to OS and ER stress. During OS, NADPH is responsible for an upregulation of miR-21 and miR-155, therefore influencing FOXO3a pathways and fibrosis [72,99]. Protein expression of CHOP can be induced and cells sensitized to apoptosis by miR-211, -689, -70, -711, -712, -762, -1897-3p, -2132, -2137 and inhibited by miR-322, -351, -503 [104,105]. OS related activation of transcription factor 6α (ATF6α) is pro-apoptotic, but can be inhibited by miR-702 [106,107]. Inhibition of miR-199a-5p results in increased ER stress-induced apoptosis [108]. In summary, both oxidative and ER stress as well as miRNAs make a decisive contribution to the development of NAFLD and the progression to NASH (summarized in Table 3). In particular, the combination of these two mechanisms provides information on pathophysiology and promises starting points for monitoring disease progression and therapy.

## 5. Viral Hepatitis

According to the WHO fact sheet, 257 million people were living with a chronic hepatits B virus (HBV) infection in 2015 with nearly 887,000 estimated deaths. Around 71 million people had a chronic hepatitis C virus (HCV) infection, resulting in an estimated 399,000 related deaths [109,110]. It has been shown that the immune system initiates the production of ROS and RNS in chronic hepatitis [111,112] and it seems that oxidative stress is important in the pathogenesis of viral hepatitis and some of these pathomechanisms are influenced by miRNAs.

Patients suffering from HCV infection produce more ROS compared to other types of virus associated hepatitis [113]. Hou et al. stated that miR-196 directly acts on Bach1 mRNA by repressing Bach1 expression and upregulating heme oxygenase 1 (HO1) leading to viral-induced oxidative stress [114]. Furthermore, miR-196 inhibits the HCV expression in HCV replicon cell lines, highlighting miR-196 as a potential therapeutic target.

As also demonstrated in other liver diseases, miR-122 also plays an important role in HCV infection. Here, miR-122 directly binds to the viral genome and enhances viral RNA replication, thus resulting in reduced miR-122 expression within the cell [115,116]. The NFκB-inducing kinase (NIK) is usually a target of miR-122, but due to the decreased levels of miR-122, NIK is increased in HCV infection [117]. In addition, HNF4α, a transcriptional regulator of miR-122 expression and known for its OS-association [118], is downregulated in HCV infection, too [117]. Both effects result in disturbance of the NIK mediated lipid metabolism and HCV-induced lipogenesis and lipid droplet formation [117,119].

Moreover, miR-122 also contributes to the pathomechanisms of HBV infection where it inhibits the effects of p53 on HBV replication by initiating a cyclin G1-p53 complex [120]. Wójcik K and co-workers described a link to oxidative stress in HVB infection as well. In a gene expression study, a positive correlation between miR-122 and NAD(P)H quinone dehydrogenase 1(NQO1) was demonstrated and it is supposed that miR-122 directly limits OS by suppression of the HBV replication and as a consequence affects the balance between pro-oxidants and antioxidants [121].

In summary, miRNAs and especially miR-122 are involved in the pathogenesis of HBV and HCV infections (see Table 4) and represent a potential target for novel treatment options [122].

## 6. Hepatocellular Carcinoma (HCC)

Hepatocellular carcinoma (HCC) is the most common primary malignancy of the liver, representing about 85% of all cases. HCC is the 6th most common malignancy worldwide and is the 4th most common cause of cancer related deaths [123]. HCC usually develops on the basis of other (chronic) liver diseases, esp. chronic viral hepatitis B or C, aflatoxin intoxication or ALD. Recently, also NAFLD and NASH became more prevalent and are now considered as major causes for HCC development in developed countries [124]. All of these conditions lead to chronic inflammation, fibrosis and cirrhosis development being essentially associated with oxidative stress conditions [125]. Interestingly, genes involved in antioxidation like Nrf2 or Keap1 were found to be mutated in up 8% of HCCs, linking the chronic stress conditions to OS pathways but also to metabolic conditions and autophagy [126], which are themselves regulated by different mechanisms, including long non-coding RNA and miRNA [127]. Under metabolic stress conditions, ROS is produced as a by-product from elevated mitochondrial fatty acid oxidation or inadequate respiratory chain function, e.g., due to fructose overload or insulin resistance. This leads to lipid accumulation which can further promote ROS production via β-oxidation of FA [125,128,129]. Additional ROS and RNS are produced by inflammatory cells that are attracted under those conditions but are also activated in case of viral hepatitis [130,131,132,133,134]. ROS can increase activity and expression of cytokines (e.g., IL-1α, IL-1β, IL-6, IL-8, TNFα) and growth factors, lead to DNA damage and trigger persistent necro-inflammation and hepatocyte regeneration that is considered a key event for HCC pathogenesis [135,136]. This can initiate a vicious circle, as the same mediators are also pathophysiologic drivers of the potentially underlying chronic liver disease, e.g., steatohepatitis, fibrosis or chronic inflammation.

8-hydroxy-2′-deoxy-guanosine (8-OHdG) was shown to be a prognostic biomarker in HCC [137]. 8-OHdG also links OS to epigenetic regulation of gene expression via DNA methylation as it is an important co-factor for the ten-eleven translocation methylcytosine dioxygenase (TET) family of DNA demethylases [138].

miRNAs have been shown to regulate expression of oncogenes and tumorsupressor genes also in HCC and provide a mechanistic link between epigenetics, inflammation, viral infection and OS [139]. Various miRNAs have been shown to be affected by OS in HCC–summarized in Table 5, e.g., downregulation of miR-26 or upregulation of miR-155 [83,140,141]. Interestingly, miR-26 expression was shown to be under the control of TET and targets the histone lysine methyltransferase Enhancer of Zeste Homolog 2 (EZH2), which is involved in the epigenetic regulation of various cell cycle control genes [142,143]. TET1 expression, in return, was shown to be under the control of miR-29b, and found to be downregulated in a study with 25 HCC patients from China [144]. In other studies, several other miRNAs, e.g., miR-494 [145] or miR-520b [146], were also shown to regulate TET1 expression in HCC, confirming the “multiple targets, multiple hits” problem and context sensitivity when analyzing miRNA signaling.

Expression of miRNA and levels of 8-OHdG were analyzed in a study comparing 29 HCC tissue samples to 58 non-cancerous liver specimens (including viral and alcoholic hepatitis). Here, significantly elevated levels of 8-OHdG were found in HCC and non-cancerous cirrhotic tissue compared to chronic hepatitis without cirrhosis or normal liver tissue. This was paralleled by increased telomerase activity and inversely correlated to telomere length. Several miRNAs were differentially regulated and the miR-17-92 cluster was down-regulated in about 50% of the analyzed samples [147]. Interestingly, the epigenetic down-regulation of miRNAs belonging to the miRNA cluster 17–92 promoted cell death in HCC cells [148]. Additional experimental findings showed that ROS reduces the expression of this miRNA cluster [149]. In HCC patients, miR-222 was found to be overexpressed and the endogenous cell cycle regulator p27^kip1^ was identified as a predicted target gene of this miRNA and expression of p27 protein is significantly decreased in HCC tissue [150]. Additionally, the tumorsuppressor is responsible for the suppression of HMGA2 leading to cell cycle block and liver cancer cell death [151].

## 7. Clinical Implications/Studies

Translating molecular scientific findings into clinical practice is the final destination of life sciences. While numberless miRNAs have been identified to play central roles in regulating nearly all pathways in cell homeostasis, it seems that science got lost in translation. OS has a key role in chronic liver diseases as it is strongly linked to acute and chronic inflammation and is therefore a main driver of progressive organ fibrosis and cancer development [152]. In chronic HCV infection antioxidant supplementation attenuates OS and although no clear clinical studies are available they are also recommended for patients with NASH [153].

Therapeutic approaches to miRNA are rare in liver diseases. Most miRNA based drugs are assessing antagonism by inhibitory antisense miRNA or by application of miRNA [154,155]. More than 6000 patents in the US market and more than 3000 in the EU market were granted in 2016 for miRNA and siRNA therapeutics [154]. Anti-miRNA oligonucleotides, so called anti-miRs or antago-miRs, have been used in experimental settings to inhibit signaling of corresponding miRNAs. Improvement of chemical structures of these oligonucleotides, e.g., adding 2′-O-methyl or 2′-O-methoxyethyl groups, generated locked nucleid acid (LNA-) antimiRs with improved pharmacokinetic and pharmacodynamic properties [156,157]. Liver specific targeting of antimiRs was achieved by conjugating these oligonucleotides to N-acetylgalactosamine (GalNAc), which is recognized by the asialoglycoprotein receptor on hepatocytes [158]. However, the therapeutic application in clinical practice seems to be far away. Actually, there are no ongoing clinical trials addressing both OS and liver disease registered to clinicaltrials.gov in a therapeutic manner.

Several trials are evaluating miRNA as biomarkers for prognosis of liver diseases–e.g., fibrosis, survival, progression of HCC. Only 10 clinical trials are registered for recruiting patients addressing microRNA and OS conditions–none of them has a therapeutic approach by addressing miRNAs.

The miR-210 group seems to be promising as a biomarker and therapeutic target in hypoxia [159]. It is up-regulated in hypoxia-related activation of HIF1α, is a key factor in induction of (tumor) cell proliferation by targeting fibroblast growth factor receptor-like 1 (FGFRL1) [160] and modulates mitochondrial alterations due to hypoxia [161]. By regulating miR-210, it could be possible to attenuate hypoxic cell damage and tissue alteration due to reperfusion after revascularization procedures. A clinical trial NCT04089943 (clinicaltrials. gov) is evaluating patients with peripheral artery disease (PAD) for the expression of miR-210 in skeletal tissue. The miR-210 group could also serve as OS marker, which could be even measured in peripheral blood [159]. By the dependency to HIF1α it could serve as prognostic factor for determining the aggressiveness and/or early stage of HCC [160,162,163].

However, OS-related miRNAs are evaluated as therapeutic target and/or biomarker for the outcome of ischemic injury such as myocardial ischemia, ischemic central insults and development of metabolic disorders. Using miRNAs as biomarkers for disease development, risk scoring, prognostic factors and drug monitoring seems actually the best approach. Countless studies are evaluating whole panels of miRNA as biomarkers in nearly all conditions of diseases (Figure 2).

### 7.1. ALD and ASH

Because of their altered expression, miRNAs could represent a valid diagnostic marker for patients affected by ALD. miR-192 and miR-30a serum levels have been correlated with ALD diagnosis [164]. Other examples are miR-103 and miR-107, which have been found to be strongly increased in the serum of patients affected by ALD and NAFLD. Their levels were low in healthy patients and in subjects affected by viral hepatitis [165]. Binge alcohol drinking caused an increase of miR-155 and miR-122 in healthy patients. Unfortunately, these miRNAs have been found over-expressed in several liver diseases and therefore could not be applied as a valid biomarker for ALD [166].

Targeting miRNAs could represent an effective therapeutic strategy for the treatment of ALD/ASH. Recently, the treatment with hyaluronic acid determined the stabilization of miR-181b-3p and importin α5 in mice fed with ethanol, thus protecting from the alcohol-derived liver damage [41]. Additionally, hyaluronic acid could normalize the level of miR-291b thus allowing the increase of Tollip and the consequent inhibition of the inflammatory pathway TLR4/NFκB [42]. Despite the contradictory role of miR-122 as determined by interrupting the cross-talk between hepatocytes and stromal cells [167], its suppression, mediated by the inhibitor Miravirsen, has shown a strong beneficial effect in chronic hepatitis suggesting a potential benefit for patients affected by ALD [167,168]. Beneficial effects for the treatment of ASH could be represented by the over-expression of liver protecting miRNAs. Unfortunately, no trial has been established to identify the clinical benefit of patients affected by ASH [168].

### 7.2. NAFLD/NASH

With rising incidence of NAFLD, obesity and diabetes in the Western and Asiatic world, NASH will be the most common cause for the development of liver cirrhosis and HCC [169]. Today, the definitive diagnosis of NASH requires a liver biopsy showing evidence with regard to steatosis, lobular or portal inflammation and ballooning of hepatocytes [170]. In NAFLD and NASH, different expression patterns of up to 44 miRNAs could be shown [78,171]. Latorre et al. and Su et al. described high serum levels of miR-451, -122, -34a and 21 in patients suffering from hepatic steatosis. miRNA-122 is elevated in the serum due to liver damage and levels are higher in severe steatosis than in mild and higher in severe fibrosis [78,79]. Furthermore, expression of miR-122 correlated positively with very low density lipoproteins (VLDL), free cholesterol and TGs [84]. With these properties miR-122 is suitable to act as a biomarker. Liu et al. was able to show that isochlorogenic acid B (ICAB) has a protective effect and is possibly associated with the ability to attenuate OS by up-regulating Nrf2 and suppressing fibrogenic factors through miR-122/HIF1α pathway [172]. Carnosic acid, an antioxidant, provides protection against NAFLD by decreasing miR-34a expression and stimulating the SIRT1/p66shc pathway [88]. In a mouse model Derdak et al. abrogated the overexpression of miR-34a with pifithrin-α *p*-nitro (PFT) and activated the SIRT1 pathway which ended up in diminished hepatic TG deposition and ameliorated the liver steatosis [91]. Kumar et al. treated mice with nanoparticles carrying a mimic of miR-29b1 which was able to significantly decrease collagen deposition in liver and serum in a liver fibrosis model. miR-29 has been associated with fibrosis in many different organs [79,173]. In a phase I trial patients suffering from fibrosis benefited from a miR-29 mimic [174]. In a transgenic mouse model overexpressing platelet derived growth factor C (PDGF C), LNA-antimiR-124 suppressed miR-124 signaling and expression of cognate target genes, leading to reduced hepatic fibrosis and even inhibited tumor formation [175]. Inhibition of miR-30b by lentiviral antimiR expression was able to reduce ER stress and improve insulin sensitivity in a high-fat dietary rat model of NAFLD [176]. These are encouraging further steps towards miRNA-directed therapies in the treatment of NASH and liver fibrosis.

### 7.3. HCC

In HCC patients, high levels of thioredoxin and manganese superoxide dismutase levels were detected and could be used as prognostic biomarkers [177,178,179]. In line with this, elevated levels of 8-OHdG, an established biomarker for oxidative stress conditions [180], were detected in various chronic liver diseases including HCV and HCC [181,182,183,184]. The miR-122 group could be another really promising candidate. It is involved in HCV related HCC progression and liver fibrosis. It targets most importantly mRNA is Aldolase A mRNA and MYC downstream regulated gene 3 [185]. Since Aldolase A is indirectly linked to hypoxia as downstream target of HIF1α [186] and its expression could be suppressed by miR-122, it could be possible to influence response to hypoxia-related survival of HCC by antagonizing miR-122. Interestingly, a nanoparticle-carrier based antimiR was able to suppress miR-122 expression for up 28 days in a murine HCV model [187]. However, in the next few years a wide range of patterns of miRNA will be available for clinical prognosis. miRNA-based drugs still need to be put into translation for clinical studies.

Besides its role in promoting tumorigenesis, OS has also been shown to exert anti-tumor effects in HCC. Downregulation or inhibition of thioredoxin reductase 1 (TXNRD1), a negative prognostic factor for HCC [188], suppressed growth of HCC models and induced sensitization to the current standard of care, sorafenib [189]. Sorafenib acts as a multi-tyrosine kinase inhibitor and impacts tumor growth by blocking the RAF/MEK/ERK pathway and by inhibiting angiogenesis [190,191]. In addition, sorafenib induced HCC cell death in vitro and in vivo also via induction of ROS production. This was linked to an increased median overall and progression free survival of patients showing higher levels of advanced oxidation protein products, which was used as a surrogate serum biomarker for OS in 26 patients [192]. Mechanistically, sorafenib blocks the mitochondrial respiratory chain and leads to disruption of the mitochondrial membranes which increases ROS production [193,194]. Resistance to sorafenib treatment is limiting its clinical efficacy. In a computational modelling approach, the miR-17-92 cluster was shown to be a key regulator of resistance to sorafenib via interaction with several components of the EGFR and IL-6 signaling pathways, including e.g., Januskinase/sterol regulatory element-binding proteins (JAK/STAT) signaling and induced myeloid leukemia cell differentiation protein (Mcl-1) function [195]. Altogether, these data confirm a complex interplay between chronic liver diseases, oxidative stress, miRNA expression, epigenetics and HCC pathogenesis.

## 8. Summary

The liver is the central metabolic organ and thus subjected to various potential external and internal factors. The increasing prevalence of NAFLD, which is projected to become the major causer of end-stage liver disease and liver transplantation, highlights the importance of understanding the pathophysiology of liver damaging conditions. While reactive (oxygen) species play a central role in normal tissue homeostasis and cellular signaling, these mediators can also contribute to acute and chronic injury of the liver, leading to fibrosis, cirrhosis and ultimately HCC formation. Recent studies demonstrated that ROS impacts lipid metabolism, detoxication, as well as central cellular survival and homeostasis processes like ER stress, calcium signaling and unfolded protein response. These pathways involve several genes that have been demonstrated to be regulated by miRNAs. While several miRNAs have now been identified to be involved in different liver diseases and some of these have been further associated to OS, we still do not fully understand the complex network of those signaling and regulatory pathways under distinct pathophysiologic conditions. Both axes, OS and miRNAs, represent potential biomarkers for surveillance, diagnosis and treatment response and may be used as novel therapeutic targets in the near future. Looking back from bed-side to bench, clinicians have to wait for stable formulations targeting miRNAs e.g., with antagomiRs enveloped into microparticles which are already available for siRNAs and being tested in clinical trials [154].

## Figures and Tables

**Figure 1 ijms-20-05266-f001:**
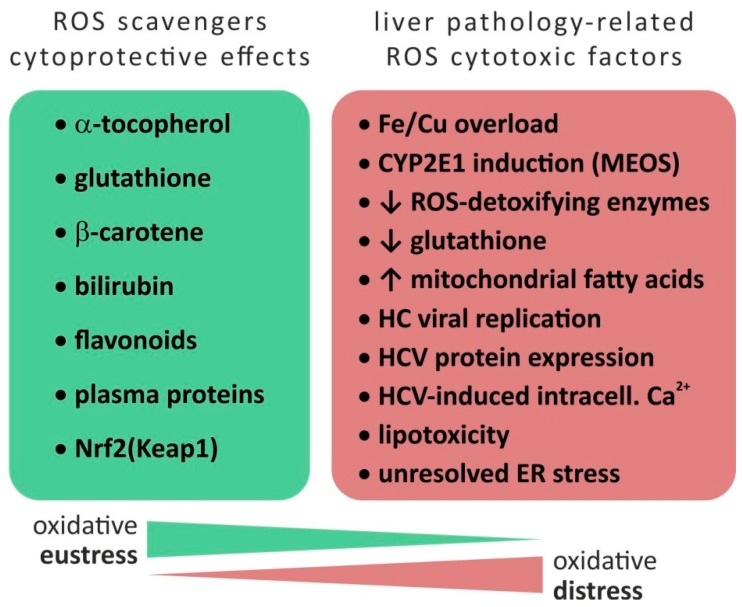
Disbalance between oxidative eustress and distress. Based on [1,22,30,33]. Abbreviations: Ca = Calcium, Cu = Copper, CYP = cytochrome P450, ER = endoplasmatic reticulum, Fe = Ferrum (Iron), GSH = Glutathione, HC = hepatitis C, HCV = hepatitis C virus, Keap1 = Kelch-like ECH-associated protein 1, MEOS = microsomal ethanol oxidizing system, Nrf2 = nuclear factor erythroid 2 like 2, ROS = reactive oxygen species, ↓ = downregulation/reduction, ↑ = upregulation/increase.

**Figure 2 ijms-20-05266-f002:**
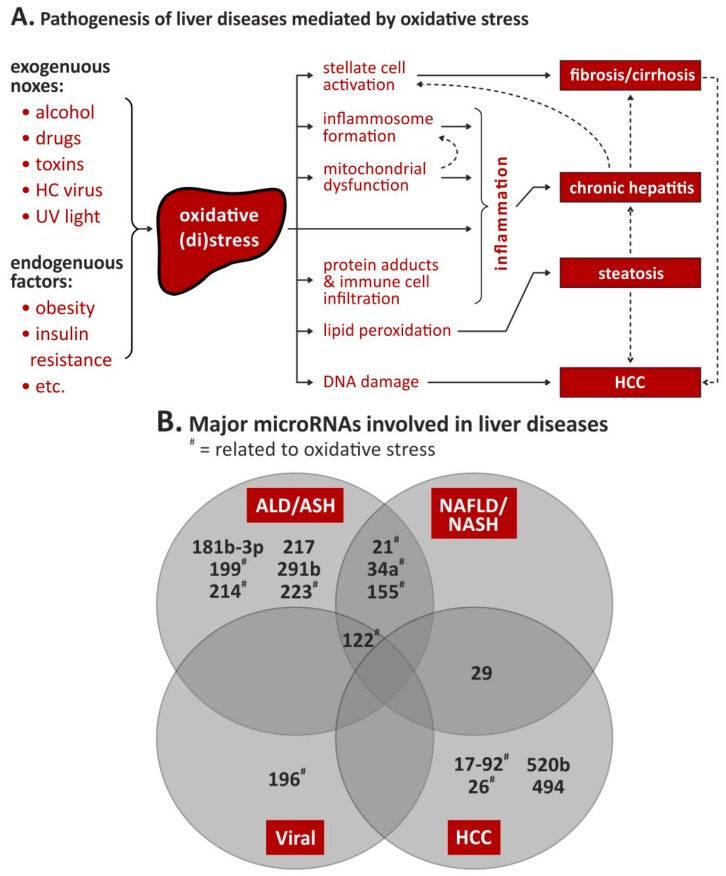
Oxidative stress- and microRNA-dependent liver pathogenesis. (**A**). Possible effects of oxidative stress and subsequent liver diseases; modified from [31]. (**B**). Venn diagram on known involvement of microRNAs in the four liver pathologies; for details, see text and Table 2, Table 3, Table 4 and Table 5. Abbreviations: ALD = alcoholic liver disease, ASH = alcoholic steatohepatitis, HCC = hepatocellular carcinoma, NAFLD = nonalcoholic fatty liver disease, NASH = nonalcoholic steatohepatitis, # = related to oxidative stress.

**Table 1 ijms-20-05266-t001:** Involvement of ROS in various liver diseases. Based on [1,22,30].

Liver Disease	ROS-Production by	(Patho)Mechanism
Hemochromatosis, Wilson’s disease	Iron/copper overload	Presence of metal catalyst for ROS production
Alcoholic liver disease (ALD)	CYP2E1 induction (MEOS)	High NADPH oxidase activity of CYP2E1 associated with production of O_2_^−^ and H_2_O_2_
	Reduced expression of ROS-detoxifying enzymes	Alcohol-induced reduction of PPARγ coactivator 1α
Nonalcoholic steatohepatitis (NASH)	Increased concentration and metabolisms of fatty acids in mitochondria	Saturation of mitochondrial β-oxidation and H_2_O_2_ production through peroxisomal β-oxidation
	CYP2E1 (CYP4A) induction	See above
HCV infection	Reduction of ROS detoxification	Reduced levels of glutathione and its regeneration as well as ROS-detoxifying enzymes
	Increased mitochondrial ROS production due to viral replication or virus protein expression	Inhibition of mitochondrial electron transport chain
	Increased NADPH oxidase triggered by calcium	Virus-induced redistribution of cellular calcium

Abbreviations: ALD = alcoholic liver disease, CYP = cytochrome P450, HCV = hepatitis C virus, MEOS = microsomal ethanol oxidizing system, NASH = nonalcoholic steatohepatitis, PPARγ = peroxisome proliferator activated receptor gamma, ROS = reactive oxygen species.

**Table 2 ijms-20-05266-t002:** Deregulated miRNAs and relation to OS in ALD/ASH.

miRNA	Evidence				Target Gene/Pathway	(Patho)Mechanism	References
	In Vitro	In Vivo	In Situ	In Silico			
155 ↑^1^	✓	✓	✓		TNFα ↑ PPARα ↓	LPS mediates the activation of NFκB. Increase of miR-155; release of TNFα, ROS and oxidative stress in Kupffer cells and hepatic stellate cells via suppression of PPARα causing overexpression of FABP4, ACC1 and LDLR	[38,39,40]
181b-3p ↑^2^	✓	✓			TLR4 ↑ NFκB ↑	Alterations in glucose and lipid homeostasis; activation of Kupffer cells	[41]
291b ↑^2^	✓	✓			Tollip ↓	Loss of downregulation of TLR4/NFκβ in Kupffer cells	[42]
34a ↑^2^	✓	✓			SIRT1 ↓	Inhibition of SIRT1 protein coding	[44]
217 ↑^2^	✓	✓			SIRT1 ↓	Alcohol-associated inflammation	[45]
122 ↓^2^		✓	✓		HIF1α ↑	miR-122 loss (deletion) or down-regulation (due to alcohol diet via GRHL2) leads to steathosis at birth, following fibrosis; miR-122 down-regulation enables HIF1α expression in ALD	[46,47]
21 ↑^2^	✓	✓			FASLG ↓ DR5 ↓	Reduced ethanol induced cell death in hepatocytes; stellate cells dysregulation via miR-21 in ethanol-induced altered extrinsic apoptotic signaling and its progression to ALD	[50]
223 ↑^1^					IL-6 ↑ p47^phox^ ↑	Peripheral neutrophils activation and liver infiltration induced by ethanol; triggering NADPH oxidase → ROS	[51]
199 ↓^1^	✓	✓			HIF1α ↑ ET-1 ↑	Leading to steatohepatitis in cirrhosis patients	[52]
214 ↑^1^	✓	✓		✓	CypP450 ↓ GSR ↓	Affecting alcohol metabolism and causing oxidative stress	[53,54]

Relation to oxidative stress: ^1^: yes, ^2^: no, ^3^: not mentioned. Abbreviations: ACC1 = Acetyl-CoA carboxylase, ALD = Alcoholic liver disease, DR5 = Death receptor 5, ET-1 = endothelin 1, FABP4 = Fatty acid binding protein 4, FASLG = Fas ligand, GRHL2 = grainyhead like transcription factor 2, GSR = glutathione reductase, HIF1α = Hypoxia Inducible Factor 1 alpha, LDLR = Low-density Lipoprotein Receptor, NFκB = nuclear factor “kappa-light-chain-enhancer”, PPARα = Peroxisome proliferator-activated receptor alpha, ROS = Reactive oxygen species, SIRT1 = Sirtuin 1, TLR4 = Toll-like receptor 4, TNFα = Tumor necrosis factor alpha, Tollip = Toll interacting protein, ↓ = downregulation/reduction, ↑ = upregulation/increase.

**Table 3 ijms-20-05266-t003:** Deregulated miRNAs and relation to OS in NAFLD/NASH.

miRNA	Evidence				Target Gene/Pathway	(Patho)Mechanism	References
	In Vitro	In Vivo	In Situ	In Silico			
21 ↑^3^	✓				PPARα ↓	Liver injury, inflammation and fibrosis	[75]
21 ↑^3^	✓	✓	✓		PTEN ↓	Development of steatosis	[73]
21 ↑^1^		✓	✓		TGFβ ↑	Induced collagen production and extracellular matrix formation fibrogenesis via increase of Col1α1 and α-SMA expression	[72]
122 ↑^2^	✓				KLF6 ↑	Activation of hepatic stellate cells and progression of liver fibrosis	[82]
34a ↑^2^	✓		✓		HNF4α ↓	Inhibition of very low-density lipoprotein secretion and promotion of liver steatosis and hypolipidemia	[89]
34a ↑^1^	✓	✓			PPARα ↓	Loss of regulation genes encoding fatty acid metabolizing enzymes and mitochondrial fatty acid oxidation activity	[87]
34a ↑^1^		✓			SIRT1 ↓	Increase of hepatic cholesterol synthesis and activation of pro-apoptotic genes (p53, p66shc)	[88]
29a and c ↓^2^	✓	✓			SIRT1 ↓	Increased levels of free cholesterol	[94,95]
29 ↑^2^	✓	✓			Col1α1 ↓	Downregulation in activated hepatic stellate cells and therefore loss of interaction with Col1α1 → decreased collagen production	[97]
155 ↑^1^	✓	✓			AKT/ FOXO3a ↑	Regulates proliferation of hepatic stellate cells promotes liver fibrosis; FOXO3a proteins maintain intracellular redox balance and survival	[99]
155 ↑^2^		✓	✓		LXRα ↓	Decreased SREBP1 and FAS resulting in an increased intracellular lipid content	[100]
155 ↑^2^		✓	✓		HIF1α and vimentin ↑	NASH-induced liver fibrosis	[101]

Relation to oxidative stress: ^1^: yes, ^2^: no, ^3^: not mentioned. Abbreviations: AKT = Protein kinase B, Col1α1 = Collagen type I alpha 1, FAS = Fatty acid synthase, FOXO3 = Forkhead-Box-Protein O3, HIF1α = Hypoxia-inducible factor 1-alpha, HNF4α = Hepatocyte nuclear factor 4 alpha, KLF6 = Krueppel-like factor 6, LXRα = Liver X receptor alpha, PPARa = Peroxisome proliferator-activated receptor alpha, PTEN = Phosphatase and Tensin homolog, SIRT1 = Sirtuin 1, SREBP1 = sterol regulatory element-binding protein, ↓ = downregulation/reduction, ↑ = upregulation/increase.

**Table 4 ijms-20-05266-t004:** Deregulated miRNAs and relation to OS in viral hepatitis.

miRNA	Evidence				Target Gene/Pathway	(Patho)Mechanism	References
	In Vitro	In Vivo	In Situ	In Silico			
196↓^1, C^	✓				Bach1/ HMOX1 ↓	Down-regulation of Bach1 gene expression, up-regulation of HMOX1 gene expression, a key cytoprotective enzyme	[114]
196↓^2, C^	✓				HCV NS5A gene ↓	miR-196 perfectly matches coding region of the HCV NS5A gene down-regulatory effect of miR-196 on HCV expression in the HCV J6/JFH1 cell culture system	[114]
122↓^2, C^	✓				HCV viral genome ↑	Enhances viral RNA replication	[115,116]
122↓^1, C^	✓	✓	✓		NIK ↑ and HNF4α ↑	Disturbance of the NIK mediated lipid metabolism → lipogenesis and lipid droplet formation → promotion of oxidative stress	[117]
122↓^2, B^			✓		cyclin G1-p53 complex ↑	Inhibits the effects of p53 on HBV replication	[120]
122↓^1, B^			✓		NQO1 ↑ and HO1 ↓	miR-122 affects balance between the pro-oxidants and antioxidants	[121]

Relation to oxidative stress: ^1^: yes, ^2^: no, ^3^: not mentioned, ^B^: Hepatitis B virus infection, ^C^: Hepatitis C virus infection. Abbreviations: HBV = Hepatitis B virus, HCV = Hepatitis C virus, HO1 = Heme oxygenase 1, HNF4α = Hepatocyte nuclear factor 4 alpha, NIK = NFκB-inducing kinase, NQO1 = NAD(P)H quinone dehydrogenase 1, NS5A = Non-structural protein 5A, ↓ = downregulation/reduction, ↑ = upregulation/increase.

**Table 5 ijms-20-05266-t005:** Deregulated miRNAs and relation to OS in liver cancer.

miRNA	Evidence				Target Gene/Pathway	(Patho)Mechanism	References
	In Vitro	In Vivo	In Situ	In Silico			
26↓^1^	✓		✓		EZH2 ↑	Sequestration of miR-26 from its target EZH2, which released the suppression on EZH2, and thereby led to EZH2 overexpression in gastric cancer	[142]
29b↓^2^	✓		✓		TET1 ↓	Feedback of miRNA-29-TET1 downregulation in HCC development suggesting a potential target in identification of the prognosis and application of cancer therapy for HCC patients	[144]
494↑^2^	✓		✓		TET1 ↓	miR-494 inhibition or enforced TET1 expression is able to restore invasion-suppressor miRNAs and inhibit miR-494-mediated HCC cell invasion	[145]
520b↓^2^	✓				TET1 ↓	Depresses proliferation of liver cancer cells through targeting 3’UTR of TET1 mRNA	[146]
17-92 cluster↓^1^			✓		E2F family ↑	ROS-mediated oxidative DNA damage correlates with over-expression of miR-92–playing a role in both the apoptotic process and in cellular proliferation pathways	[147]

Relation to oxidative stress: ^1^: yes, ^2^: no, ^3^: not mentioned. Abbreviations: E2F = E2F transcription factor family, EZH2 = Enhancer of zeste homolog 2, TET1 = Ten-eleven translocation methylcytosine dioxygenase 1, ↓ = downregulation/reduction, ↑ = upregulation/increase.

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
