# Peer review of "The Crosstalk of miRNA and Oxidative Stress in the Liver: From Physiology to Pathology and Clinical Implications"

_ijms, 2019, doi:10.3390/ijms20215266_

Round 1

Reviewer 1 Report

The manuscript titled “The crosstalk of miRNA and oxidative stress in liver: from physiology to pathology and clinical implications” by Klieser et al. presents a review on the physiological and pathological importance  of oxidative stress in liver, mainly focusing on the regulatory role of miRNAs. As an ever expanding field, aberrant oxidative stress has been implicated in multiple biological processes as well as malignancies. Unlike huge existed review papers talking about their role in liver diseases, the authors chose to summarize the crosstalk between miRNAs and oxidative stress, which is an interesting topic. A few suggestions:

Anti-miRNA oligonucleotides (2′-O-Methyl AMOs; 2′-O-Methoxyethyl AMOs; Locked nucleic acid AMOs) could be briefly discussed to address the potential of miRNA-based therapeutics in liver diseases. A cartoon could be added to help explain the generation of oxidative stress caused by alcohol, drugs and other factors. Many long sentences are hard to understand which should be split. Keep the abbreviation constant throughout the draft. “miRNA” or “miRNAs”? Several linguistic problems. A few examples:

Line 27, “exposition” . Do you mean “exposure”?

Line 258, “MiRNA-155 is upregulated in mouse models inducing NASH via high fat diet”. Do you mean” MiRNA-155 is upregulated in a NASH mouse model induced via the high fat diet”?

Line 399-400, “there a no trials”

Author Response

Revision of manuscript ijms-618571 „The crosstalk of miRNA and oxidative stress in liver: from physiology to pathology and clinical implications” by Klieser et al.

Dear editor

We thank the editor and the reviewers for their comments on our manuscript and the chance to revise it according to the reviewers’ suggestions.

All changes are marked in track change and are incorporated in the revised manuscript file. Please find a detailed point-by-point reply to the reviewers’ comments below.

We think that all comments have been adequately addressed and are looking forward to receiving your positive feedback on our manuscript.

Yours sincerely,

Eckhard Klieser on behalf of the co-authors

Reviewer 1:

We thank the reviewer for his positive feedback and critical suggestions.

We added a brief discussion on antimiRs in chapter 7 and subsequent subchapters.

The language of the manuscript was carefully reevaluated and minor changes were implemented throughout the manuscript text (e.g. shortening of sentences, consistent use of abbreviations).

We have corrected lines 27 and 258 as suggested. The sentence in line 399-400 was deleted as it is redundant with the paragraph before.

An additional Figure 2 was included in chapter 7 to summarize the discussed findings and connections between oxidative stress and miRNAs in the liver.

Reviewer 2 Report

The manuscript “The crosstalk of miRNA and oxidative stress in liver: from physiology to pathology and clinical implications”, describes the changes at the miRNA level, the formation and impact of ROS and RNS and the crosstalk between several factors in the development and progression of acute and chronic liver diseases.

The work has clinically important data still some minor points need to be adjusted before publication.

Main points:

A general check to the format of the manuscript is required, since there are several parts with different line space, fonts; Figure 1: it would be better if the font was the same as the rest of the manuscript. Also, the figure is slightly cut. Check this; All tables: correct the font (same as the rest of the manuscript); Lines 109, 144, 149 and other cases: correct MiRNA- to miRNA-; Check the abbreviations: use them after defining the abbreviations (e.g. HCC); Try to change the references from the 90’s into more recent ones; More important: this reviewer thinks that the review would improve if the explanation of the diseases would not be so extensive. In fact, it is very difficult to maintain the full attention with so many details along the text. Try to select and summarize the most important facts that link miRNAs and liver disease.

Author Response

Revision of manuscript ijms-618571 „The crosstalk of miRNA and oxidative stress in liver: from physiology to pathology and clinical implications” by Klieser et al.

Dear editor

We thank the editor and the reviewers for their comments on our manuscript and the chance to revise it according to the reviewers’ suggestions.

All changes are marked in track change and are incorporated in the revised manuscript file. Please find a detailed point-by-point reply to the reviewers’ comments below.

We think that all comments have been adequately addressed and are looking forward to receiving your positive feedback on our manuscript.

Yours sincerely,

Eckhard Klieser on behalf of the co-authors

Reviewer 2:

We thank the reviewer for his positive evaluation of our manuscript.

Manuscript formatting has been adapted according to the journal’s style requirements.

The formatting of “MiRNA” has been corrected to “miRNA”. Distinct miRNAs are now referred to as “miR-XXX” instead of “miRNA-XXX”.

The use of abbreviations was checked and adapted throughout the manuscript.

Several references have been added but older key papers still remain to be cited.

The introduction to chapters 4 (NAFLD and NASH) and 6 (HCC) was shortened while the other introductory remarks are considered to be concise and focused.

Reviewer 3 Report

The manuscript is focused on "The crosstalk of miRNA and oxidative stress in liver: from physiology to pathology and clinical implications". 

In my opinion, the manuscript has a significant contribution to the subject. Hence, I recommend publishing the manuscript. 

However, the English style and grammar should be improved before publication. In order to increase interest among the readers, I also recommend a significant improvement of the illustrative layer of the paper, by adding some diagrams and schemes. 

Author Response

Revision of manuscript ijms-618571 „The crosstalk of miRNA and oxidative stress in liver: from physiology to pathology and clinical implications” by Klieser et al.

Dear editor

We thank the editor and the reviewers for their comments on our manuscript and the chance to revise it according to the reviewers’ suggestions.

All changes are marked in track change and are incorporated in the revised manuscript file. Please find a detailed point-by-point reply to the reviewers’ comments below.

We think that all comments have been adequately addressed and are looking forward to receiving your positive feedback on our manuscript.

Yours sincerely,

Eckhard Klieser on behalf of the co-authors

Reviewer 3:

We thank the reviewer for his positive evaluation and support of our manuscript.

An additional figure 2 was included in chapter 7, see also request from Reviewer 1, to further illustrate oxidative stress in the liver.

The language of the manuscript was carefully reevaluated and minor changes were implemented throughout the manuscript text (e.g. shortening of sentences, consistent use of abbreviations).